# Template-Free Preparation of a Mesopore-Rich Hierarchically Porous Carbon Monolith from a Thermally Rearrangeable Polyurea Network

**DOI:** 10.3390/ijms23084271

**Published:** 2022-04-12

**Authors:** Junsik Nam, Yusin Pak, Gun Young Jung, Ji-Woong Park

**Affiliations:** School of Materials Science and Engineering, Gwangju Institute of Science and Technology, 123 Cheomdangwagi-ro, Buk-gu, Gwangju 61005, Korea; namjunsik@gist.ac.kr (J.N.); rhpys@gist.ac.kr (Y.P.); gyjung@gist.ac.kr (G.Y.J.)

**Keywords:** polyurea network, hierarchical pore, porous carbon, carbon monolith, moldable carbon

## Abstract

A mesopore-rich, hierarchically porous carbon monolith was prepared by carbonizing a polyisocyanurate network derived by thermal rearrangement of a polyurea network. The initial polyurea network was synthesized by the cross-linking polymerization of tetrakis(4-aminophenyl)methane (TAPM) and hexamethylene diisocyanate (HDI) in the sol-forming condition, followed by precipitation into nanoparticulate solids in a nonsolvent. The powder was molded into a shape and then heated at 200–400 °C to obtain the porous carbon precursor composed of the rearranged network. The thermolysis of urea bonds to amine and isocyanate groups, the subsequent cyclization of isocyanates to isocyanurates, and the vaporization of volatiles caused sintering of the nanoparticles into a monolithic network with micro-, meso-, and macropores. The rearranged network was carbonized to obtain a carbon monolith. It was found that the rearranged network, with a high isocyanurate ratio, led to a porous carbon with a high mesopore ratio. The electrical conductivity of the resulting carbon monoliths exhibited a rapid response to carbon dioxide adsorption, indicating efficient gas transport through the hierarchical pore structure.

## 1. Introduction

Carbon is critical in catalysis, energy storage and conversion, and molecular separation. Fabricating carbon with a hierarchical pore structure is essential for rapid molecular transport between the inner pore surface and the bulk medium [1,2,3,4,5,6]. The hierarchical pore structure consists of macropores (>50 nm), mesopores (2 nm < d < 50 nm), and micropores (<2 nm) that are three-dimensionally interconnected [7]. The microporous structure gives a high specific surface area for the active sites, and the macropores are responsible for the molecular transport of a larger volume. Meanwhile, it must contain a sufficient ratio of mesopores to facilitate molecular diffusion between the macropores and micropores [8,9]. In general, the micropores are developed in typical carbonization processes, and the macropores can be produced by employing the morphology of the carbon precursors that are greater than tens of nanometers. In contrast, the formation of mesopores is often insufficient in the usual carbonization of organic materials [10,11]. New pre-carbon materials that provide structures with mesopores, as well as micropores and macropores, should enable the exploitation of valuable properties of carbon [12,13].

Many approaches to introduce mesopores into carbonaceous material employed inorganic mesoporous templates [14,15]. However, this requires chemical etching processes to remove the template [16,17,18], which often result in undesired chemical structure formation, or the deformation or collapse of the pore structure. In addition, the problems in the uniformity of the pore structure may become severe when the material synthesis is scaled up.

Pore templates may be avoided if a carbon precursor forms a pore structure that is sufficiently stable until carbonized at elevated temperatures. In this regard, thermally rearrangeable polymers, consisting of a thermostable polymer matrix with thermally decomposable segments, are ideal candidates [19,20,21]. The labile segments are dissociated and evaporated, generating pores within the polymer matrix in the temperature range before carbonization. 

The polyurea network (UN) [22,23], prepared from cross-linking polymerization of an aromatic tetra-amine and an aliphatic diamine, has been reported to show thermal rearrangement chemistry, comprising the dissociation of the urea bonds to amines and isocyanates, and the trimerization of isocyanates into cyclic isocyanurate rings [24,25,26]. During the process, micro- and mesopores are generated via the expulsion of aliphatic moieties. The rearranged polyurea network (RUN) contains many isocyanurate nodes that are resistant to high temperatures up to 500 °C.

In addition, the UN can be produced in a nanoparticulate form using its sol-gel processability. Thanks to their thermal rearrangement chemistry, the powders may be molded and sintered in a designed shape. The sintering processability of the UN and RUN may allow carbon to be molded in a non-powdery form if they can be carbonized successfully. This may allow binders or solvents to be avoided, which would cause pore blockage and decrease the surface area [27,28], to produce a non-powdery carbon object. The nanoparticulate morphology of the carbon precursors with thermal sintering ability also provides a facile route to the macroporous structure in their carbonized product.

We report the synthetic methods for mesopore-rich, hierarchical porous carbon monoliths from polyurea networks. First, the thermal treatment conditions for the UN were varied to obtain the RUN. Next, the chemical structural details and the pore characteristics of the RUN were investigated. The RUNs were then carbonized, and the porosities of the resultant carbon were analyzed. As a result, the carbon monoliths exhibited a higher mesopore ratio when obtained from the RUN with a high isocyanurate content, suggesting that the presence of thermostable isocyanurate nodes is crucial for the generation of mesopores in the carbonized RUN (CRUN). Lastly, the carbon monolith exhibited significant electrical conductivity, which responded rapidly to the carbon dioxide adsorption and desorption on its pore surface. The rapid conductivity response to switching the atmospheric gas molecules was ascribed to the hierarchical porosity of the carbon monolith.

## 2. Materials and Methods

### 2.1. Materials

TAPM was prepared by the previously reported method [29]. HDI (99%, Sigma-Aldrich, St. Louis, MO, USA) was freshly distilled under reduced pressure. Anhydrous N, N-dimethylformamide (DMF) (99.8%, Sigma-Aldrich, St. Louis, MO, USA) was used without further purification.

### 2.2. Synthesis of CRUNs

In a typical run for the synthesis of CRUN, TAPM (0.17 mmol and 0.064 g) and the HDI (0.34 mmol and 0.056 g) were dissolved in 1.59 mL and 1.41 mL of DMF, respectively. The TAPM solution was drop-wisely added to the HDI solution at room temperature under a nitrogen atmosphere. The mixture was stirred for 60 h to yield UN sol. The UN sol was precipitated in copious amounts of acetone. The precipitate was washed three times with acetone. The powdery solid was isolated by filtration and dried for 48 h at 150 °C in a vacuum oven (VOS-310C, Sunil Eyela, Sungnam, Korea). A UN monolith was prepared by pressing 100 mg of the UN powder. The UN monolith was thermally treated in a programmable muffle furnace (Daihan FX-27, Daihan Scientific, Wonju, Korea), yielding the RUN monolith. Each UN monolith was heated to 300, 320, 340, and 360 °C at a rate of 2 °C min^−1^ under a nitrogen atmosphere, kept at the final temperature for 12 h, and then cooled rapidly to room temperature. The resulting RUN monoliths were denoted as RUN300, RUN320, RUN340, and RUN360, depending on the final temperature. The RUN monoliths were further carbonized in a programmable tube furnace (Daihan FT830, Daihan Scientific, Wonju, Korea), yielding the CRUN monoliths. All RUN monoliths were heated to 800 °C at a rate of 2 °C min^−1^ under a nitrogen atmosphere, kept at the final temperature for 1 h, and then cooled rapidly to room temperature. The resulting CRUN monoliths were denoted as CRUN300, CRUN320, CRUN340, and CRUN360.

### 2.3. Characterizations

Scanning electron microscope (SEM) images were collected on SU 70 (Hitachi, Tokyo, Japan). The SEM samples were dried in a vacuum for 24 h and sputtered with platinum for 60 s using the ion sputter (E-1030, Hitachi, Tokyo, Japan). Transmission electron microscope (TEM) images were collected on Tecnai F20 ST (FEI, Hillsboro, OR, USA) for the 100 nm-thick sections prepared by the focused ion beam (FIB) using Versa 3D DualBeam (FEI, Hillsboro, OR, USA). X-ray photoelectron spectroscopy (XPS) experiment was conducted with ESCALAB 250XI (Thermo Fisher Scientific, Waltham, MA, USA), equipped with a 14.5 kV energy source in an ultrahigh vacuum condition. Nitrogen adsorption–desorption isotherms were recorded on an ASAP 2020 volumetric adsorption apparatus (Micromeritics, Norcross, GA, USA) at 77 K. Before analysis, the samples were degassed in the degassing port of the adsorption analyzer at 473 K for at least 12 h. The surface area, pore volume, and pore size distribution were calculated using ASAP 2020 v3.00 software. Fourier transform infrared (FT-IR) spectroscopy was carried out on a Nicolet iS10 (Thermo Fisher Scientific, Waltham, MA, USA) for the samples in KBr pellets. The electrical conductivity of the carbon, in response to CO_2_ adsorption, was measured on a homemade apparatus, as shown in the Appendix A.

## 3. Results and Discussion

The carbon monolith was prepared by the procedure illustrated in Figure 1. The polymerization of TAPM and HDI in DMF (Figure 1a) maintained homogeneous sol until the polymerization mixture reached the gelation time (*t_g_*) (~80 h) [30,31,32]. The UN was obtained in nanoparticulate forms, by precipitating the polymerization mixture into acetone approximately 60 h after starting the polymerization. The SEM image of the UN powder shows the nanoparticulate morphology (Figure 1a).

The powder was pelletized into a disk shape under pressure and treated thermally in two steps, for rearrangement and carbonization, respectively (see Experimental section for details). Optical photographs of the powdery UN and the disk-shaped UN, RUN, and CRUN are shown in Figure 1a. The initial UN disk was brittle, but hardened after rearrangement by heating at 200 to 400 °C. It is known that the urea bonds dissociate into amines and isocyanates, and the isocyanates cyclize to form isocyanurate (Figure 1b) in this temperature range [33,34]. We varied the rearrangement temperature and time, and then analyzed the chemical structure and porosity of the resultant RUNs. Once the rearrangement process finished, the resultant network was heated slowly to a carbonization temperature to obtain the CRUNs.

The thermogravimetric analysis (TGA) curve of the UN shows about 40% weight loss, corresponding to the weight fraction of the hexamethylene moiety, in the range of 300 to 400 °C (Appendix A). FT-IR confirmed the chemical transformation from the UN into the RUN. The isocyanurate peak appeared near 1710 cm^−1^ in the RUNs, with the intensity varying with the rearranging temperature (Figure 2a) [25]. Alkyl moieties were removed nearly entirely in the samples treated above 340 °C, as indicated by the disappearance of C-H stretching bands at 2930 and 2860 cm^−1^. Nevertheless, the intensities of the isocyanurate peak at 1710 cm^−1^, relative to a phenyl C=C stretching peak at 1605 cm^−1^, varied with the rearrangement temperature, indicating that the thermal conditions determine the ratio of isocyanurate rings in the rearranged network. As judged from the IR spectra, RUN340, obtained by heating the UN at 340 °C for 12 h, contained the highest ratio of isocyanurate, while completely removing the alkyl moieties.

XPS for the RUN gave consistent results with FT-IR. The N1’s peak appeared as a single peak at 400.1 eV in the UN, corresponding to the urea nitrogens. In contrast, an additional N1 peak appeared at 398.4 eV in RUN340, corresponding to the isocyanurate nitrogens (Figure 2b and Appendix A) [35]. A quantitative analysis was performed to investigate the ratio of isocyanurate in the RUN, according to the rearrangement temperature and time (Figure 2c and Appendix A). The UN sample was heated to a final temperature of 300, 320, 340, or 360 °C, and then remained there for 1, 12, or 24 h. Figure 2c,d show that the isocyanurate nitrogen ratio of the RUN reached the maximum when obtained by heating at 340 °C for 12 h. Heating below 340 °C resulted in a low isocyanurate ratio, with incomplete removal of the alkyl moieties. Heating higher than 340 °C also reduced the isocyanurate content, likely caused by the facilitated removal of isocyanate-bearing groups at temperatures that were too high [36].

The porosities of the UN and RUN were measured from the nitrogen adsorption–desorption isotherms shown in Figure 3a. The desorption isotherms showed different hysteresis types for the UN and RUN; the UN showed H1 hysteresis at high relative pressure, and the RUNs (except for RUN300) showed H4 hysteresis, indicating that the thermal rearrangement process developed micropores and mesopores. The non-linear density functional theory (NLDFT)-based pore distribution curves are shown in Figure 3b. The UN mainly possessed pores in the macropore range, likely resulting from the nanoparticulate morphology. RUN300 lacked porosity, which may be accounted for by the loss of the gaps between the particles via particle sintering, through reversible dissociation and reformation of urea bonds. The formation of isocyanurate rings and the evaporation of alkyl moieties must have been too slow to fix the initial morphology at the low rearrangement temperature. RUN320, RUN340, and RUN360 exhibited nitrogen adsorption in the range of low relative pressures (<0.1 P P_0_^−1^). This result and the data in Figure 2a show evidence that the microporosity is developed by the urea bond rearrangement, accompanied by the evaporation of hexamethylene moieties [25,37,38]. The RUN340 and RUN360 have exceptionally high volumes of mesopores lower than 10 nm and micropores near 2 nm than other samples (Figure 3b).

The RUNs were carbonized by heating to the final temperature of 600, 700, 800, or 900 °C, and remained for 1 h. The mesopore ratio of the CRUN from RUN340 reached the maximum when carbonized at 800 °C. In the meantime, the electrical conductivity increased with the carbonization temperature (Appendix A). Since we aimed to produce carbon with a high mesopore volume, we set the carbonization condition to 800 °C (see Appendix A for the porosity and electrical conductivity of the CRUNs in different carbonization conditions). 

Figure 3c,d show the nitrogen adsorption–desorption isotherms and the NLDFT pore distribution curves for the CRUNs from RUNs 300, 320, 340, and 360. The sorption isotherms of CRUNs showed type H4 hysteresis, similar to the RUNs. The pore size of the CRUNs is distributed over a wide range between subnanometer to 100 nm, suggesting that the carbon monoliths possessed hierarchical porosity. Direct carbonization of the UN without the rearranging step resulted in negligible porosity in the micro- and mesopore range, demonstrating that the formation of a stable pore structure of the carbon precursor in the intermediate temperature range was critical. CRUN340 showed the highest mesopore volume (0.065 cm^3^/g) and surface area ratio (11%) among the CRUNs (Appendix A). CRUN360 showed a lower micro- and mesopore volume than CRUN340, which contrasts with the higher microporosity in its precursor RUN360. This result indicates that the higher content of thermo-resistant isocyanurate nodes in RUN340 than in RUN360 prevented the collapse of pore structure during the carbonization process. 

The electron microscopic images (Figure 4) show that the hierarchical pore structures developed in the transformation of the UN to RUN, and then to CRUN, consistently with the porosity analysis. The images also show that the CRUN is a monolithic solid. The SEM images of the UN, RUN, and CRUN clearly show the gap between the nanoparticulate UNs developed to the macropores in the RUN and CRUN through sintering of the particles (Figure 4a–c). The TEM images for the ultrathin sections of the RUN and CRUN show macropores with shapes reminiscent of the gap between spherical nanoparticles. The development of mesopores is also visible in the TEM image of the RUN340 and CRUN340 (Figure 4d,e insets), which are distinct from the image of non-porous UN (Appendix A). RUN340 and CRUN340 showed irregular, but interconnected, pores of a few nanometers in size.

Based on the above results, the mechanism for creating hierarchical pores in CRUN is proposed, as depicted in Figure 5. The molded UN only has macropores in the gaps between the particles (Figure 5a). The macropores are maintained in the sintering and carbonization processes. Micropores and mesopores are developed via the evaporation of alkyl moieties during the thermal rearrangement into RUN (Figure 5b). The rearrangement process generates the RUN rich in thermo-stable isocyanurate nodes. The isocyanurate-rich RUN sustains the high carbonization temperature, although the pore sizes increased via further loss of weak organic moieties, giving the CRUN. The micropores are additionally generated via the formation of amorphous carbon [36].

The bulk density of CRUN340 was 0.53 g/mL, as measured by mercury intrusion porosimetry. The CRUNs exhibited high-capacity carbon dioxide adsorption of 4~5 mmol/g (Appendix A). The high affinity to carbon dioxide may be attributed to the large pore volume and the highly developed hierarchical porosity. In addition, elemental analyses indicated the presence of nitrogen and oxygen in significant weight fractions in the CRUNs (Appendix A), which should enhance the materials’ affinity to carbon dioxide. Despite the high heteroatom contents, the carbon monoliths exhibited electrical conductivity of about 1 S/cm.

The CRUN monoliths’ excellent carbon dioxide adsorption and electrical conductivity were utilized to investigate how their porosity affects their gas transport rates. We monitored the current change in the carbon monolith occurring with carbon dioxide adsorption and desorption [39]. The experimental setup is given in Appendix A. As shown in Figure 6, the current increased upon purging carbon dioxide into the cell and decreased upon purging nitrogen. Three CRUN samples, with different porosities (Appendix A), were employed in the test. CRUN 340 exhibited the biggest and fastest current change in response to the change in atmosphere. While the amount of current change should reflect the total surface area of the CRUN, the rate of current change must be related to the rate of gas transport through the pore structure. It appeared that the average times to current saturation after switching to carbon dioxide were 143, 108, and 278 s for the CRUN320, CRUN340, and CRUN360, respectively. The results indicate that the hierarchical porosity, with a high mesopore ratio, of the CRUN340 facilitated gas diffusion.

## 4. Conclusions

In summary, we synthesized a mesopore-rich, hierarchical porous carbon monolith by carbonizing thermally stable polyisocyanurate networks prepared by the rearrangement of polyurea networks. The initial polyurea network was synthesized by cross-linking polymerization of TAPM and HDI in the sol-forming condition, followed by isolation as powders of nanoparticulate solids. The UN was pressure molded and then heated between 200 and 400 °C, yielding the RUN as a monolith containing isocyanurate nodes, micropores, and macropores. The rearranged network with a high isocyanurate ratio was heated further to transform it into a hierarchically porous monolith of CRUN, with a high mesopore ratio. The electrical conductivity of the CRUN monoliths exhibited a rapid response to carbon dioxide adsorption, indicating facile gas transport through the hierarchical pore structure. The current method, developed to prepare hierarchically porous carbon from a thermally rearrangeable polyurea network, is promising for synthesizing various carbonaceous materials with molecular adsorption, separation, and catalytic functionalities.

## Figures and Tables

**Figure 1 ijms-23-04271-f001:**
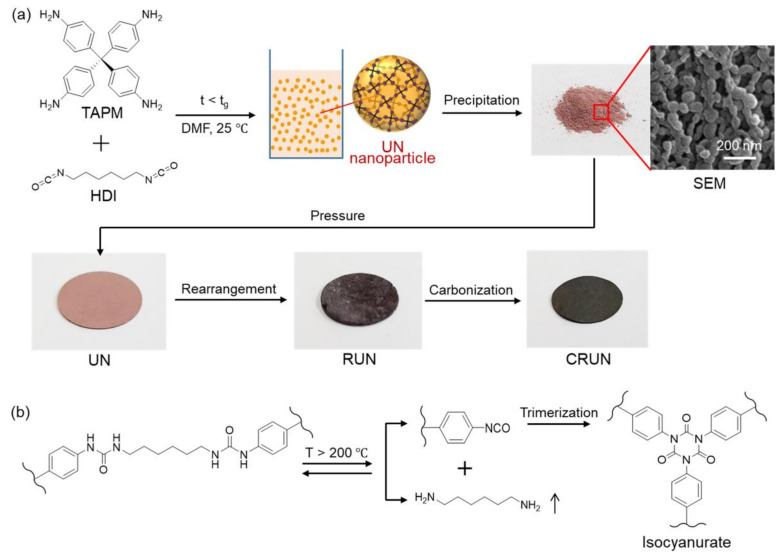
Schematic illustration of the preparation of the UN and its transformation into RUN and CRUN. (**a**) The polymerization of TAPM and HDI to obtain the UN in the sol state. Optical photographs of the powdery UN were obtained by precipitating the UN sol into acetone, the disk-shaped UN (diameter 1.3 cm and thickness 0.5 cm), RUN (diameter 1.1 cm and thickness 0.4 cm), and CRUN (diameter 0.9 cm and thickness 0.3 cm). The SEM image shows the nanoparticulate morphology of the UN powder. (**b**) A major change in chemical structure occurs with the rearrangement of urea networks.

**Figure 2 ijms-23-04271-f002:**
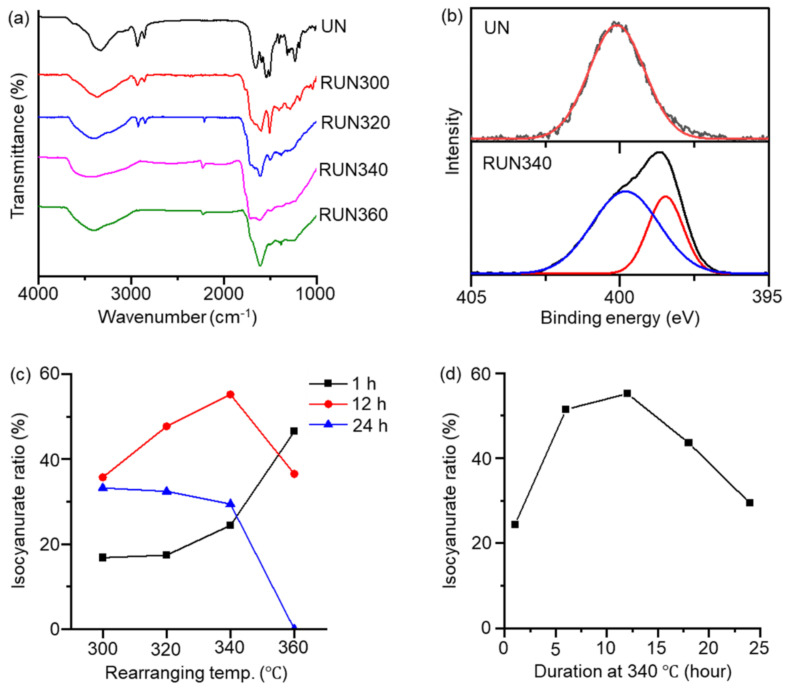
Chemical compositions of the RUNs varying with the thermal treatment conditions. (**a**) FT-IR spectra of the RUNs treated at different final temperatures for 12 h. (**b**) XPS N1’s spectra of the UN and RUN340. (**c**) Isocyanurate ratio of RUNs prepared with different final temperatures for durations of 1, 12, and 24 h. (**d**) Isocyanurate ratio of RUN samples prepared for different durations at 340 °C.

**Figure 3 ijms-23-04271-f003:**
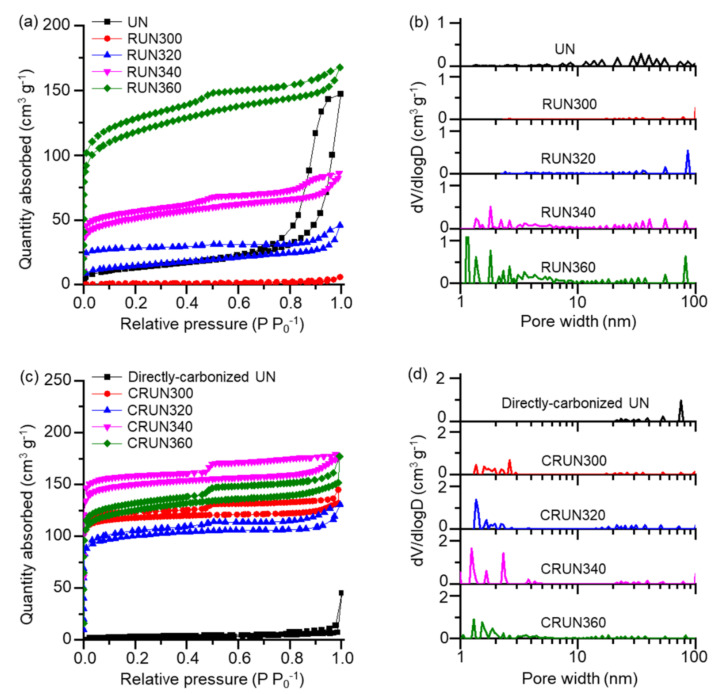
The change in pore characteristics in the transformation of RUNs to CRUNs. (**a**) N_2_ adsorption–desorption isotherms of the RUNs and a UN. (**b**) NLDFT pore distribution curves of the RUNs and a UN. (**c**) N_2_ adsorption–desorption isotherms of the CRUNs and a directly carbonized UN. (**d**) NLDFT pore distribution curves of the CRUNs and a directly carbonized UN.

**Figure 4 ijms-23-04271-f004:**
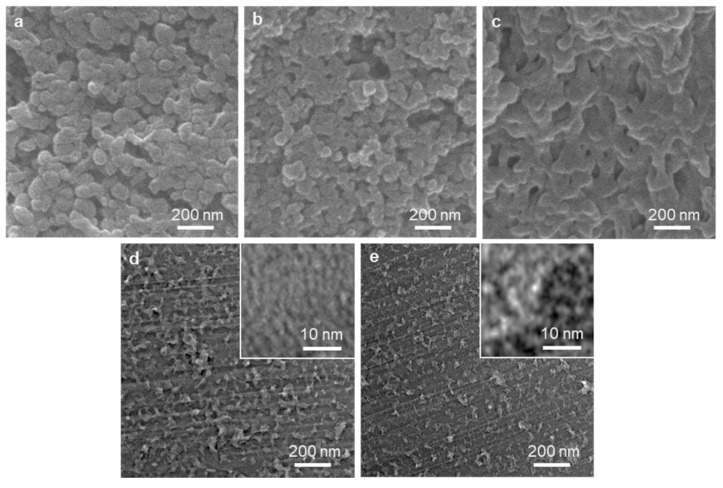
Structural characterization of UN, RUN340, and CRUN340. SEM images of the cross-section of (**a**) UN, (**b**) RUN340, and (**c**) CRUN340. TEM image of the cross-section of (**d**) RUN340 and (**e**) CRUN340.

**Figure 5 ijms-23-04271-f005:**
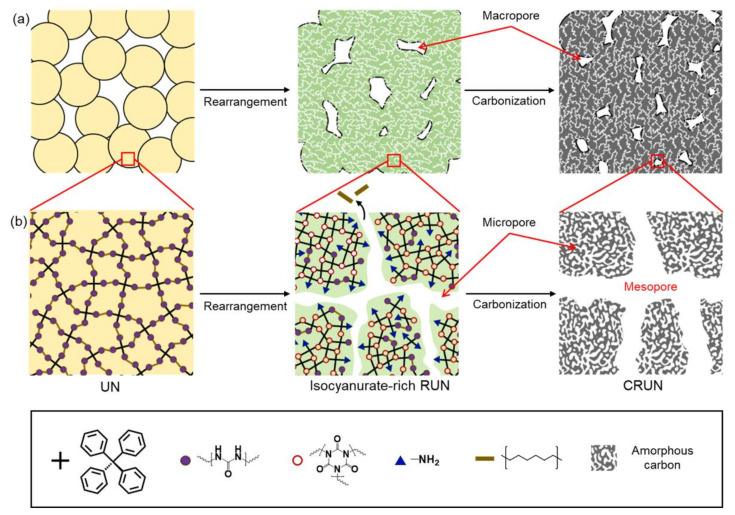
Schematic illustration of pore formation and structural change in the transformation from UN to RUN to CRUN. (**a**) Macropore formation in UN, RUN, and CRUN. (**b**) The formation of the mesopore and micropore with the change in network structure in the UN, RUN, and CRUN.

**Figure 6 ijms-23-04271-f006:**
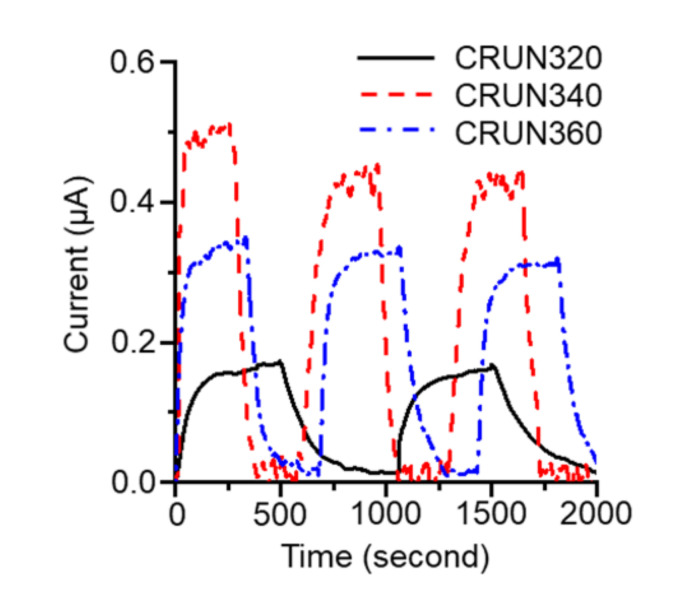
The current change in CRUNs occurred by switching the purge gas between carbon dioxide and nitrogen. See Appendix A for the experiment details.

## Data Availability

Not applicable.

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
