# Peer review of "Template-Free Preparation of a Mesopore-Rich Hierarchically Porous Carbon Monolith from a Thermally Rearrangeable Polyurea Network"

_ijms, 2022, doi:10.3390/ijms23084271_

Round 1
Reviewer 1 Report
The authors developed polyisocynuarate networks which can be developed into hierarchical porous networks. The synthesis is quite easy and straightforward, and the rearrangement of networks generated porosities, which was systematically studied and confirmed by N2 adsorption-desorption, IR and XPS analysis. The reviewer only have some minor comments for potential improvements before publishing:
- It would be nice to include any elemental composition analysis (either from XPS or combustion based elemental analysis), since this might be related to the CO2 adsorption and transportation.
- In Fig. 3, why did the authors use NLDFT-calculated pore size distribution instead of some other common methods, such as BET? And for Fig. 3a/3c, it would be nice to discuss the different hysteresis shapes for UN and RUN networks/carbons.
- Just a suggestion, the authors might also do mechanical analysis on the UN vs. RUN to see if there are any changes in the modulus. But it is not strictly necessary for this manuscript.
Author Response
Reviewer 1
The authors developed polyisocyanurate networks which can be developed into hierarchical porous networks. The synthesis is quite easy and straightforward, and the rearrangement of networks generated porosities, which were systematically studied and confirmed by N2 adsorption-desorption, IR, and XPS analysis. The reviewer only has some minor comments for potential improvements before publishing:
(1) It would be nice to include any elemental composition analysis (either from XPS or combustion-based elemental analysis) since this might be related to the CO2 adsorption and transportation.
- The elemental analysis data for four CRUNs in Table R1 below was added in the supplementary information (Table S3). The CO2 adsorption curves were also obtained and provided in the supporting info. (Figure S9). We added a new paragraph discussing the carbon dioxide adsorption behavior in the main text of the revision (lines 250~256)
Table R1. Elemental composition of CRUNs.
Material |
C (wt%) |
N (wt%) |
O (wt%) |
H (wt%) |
CRUN300 |
76.32 |
7.21 |
14.75 |
1.71 |
CRUN320 |
75.01 |
7.36 |
15.71 |
1.91 |
CRUN340 |
75.56 |
7.58 |
14.93 |
1.93 |
CRUN360 |
78.09 |
5.31 |
14.80 |
1.79 |
(2) In Fig. 3, why did the authors use NLDFT-calculated pore size distribution instead of some other common methods, such as BET? And for Fig. 3a/3c, it would be nice to discuss the different hysteresis shapes for UN and RUN networks/carbons.
- The BET method is suitable for surface area calculations. The reviewer probably refers to the Barrett-Joyner-Halenda (BJH) pore distribution analysis. The BJH pore distribution is usually obtained from the desorption part of the isotherms. Oftentimes artifacts occur because of the cavitation effect in the range of pore diameter of 2-5 nm. The NLDFT method avoids the cavitation artifact because it employs the adsorption part of the isotherms.
- We added short discussions on the hysteresis of the isotherm curves for UN, RUN, and CRUNs in the revision (lines 177-180, and 201-202).
(3) Just a suggestion, the authors might also do a mechanical analysis on the UN vs. RUN to see if there are any changes in the modulus. But it is not strictly necessary for this manuscript.
- Thank you for the suggestion. Unfortunately, we could not measure mechanical properties because it was difficult to prepare specimens for standard mechanical tests. Qualitatively speaking, the RUN becomes harder than the UN.

Reviewer 2 Report
The authors presenting an interesting work to synthesize hierarchically porous carbon monolith using a thermally rearrangeable PU network. I think some of the important issues should be provided.
(1) the porous carbon monolith is mesopore-rich, waht is the percentage of mesopores?
(2) What's happend during the reaggrange process when the UN was heated at 300, 320, 340, and 360 oC?Did the UN under go thermal degradation? Thus TGA was suggested to provided in the revised manuscript.
(3) what's the density and porosity of the carbon monolith?Could they compressiable?
(4) Figure 1 must be revised, there is a vertical line in the left.
Author Response
Reviewer 2
The authors present an interesting work to synthesize hierarchically porous carbon monolith using a thermally rearrangeable PU network. I think some of the important issues should be provided.
(1) the porous carbon monolith is mesopore-rich, what is the percentage of mesopores?
- We estimated the percentage of mesopores by the surface area ratio of the CRUNs (Table S2) in the supplementary information. The mesopore surface area ratio over BET surface area of CRUN300, CRUN320, CRUN340, and CRUN360 were 4, 7, 11, and 5%, respectively.
(2) What's happened during the rearrange process when the UN was heated at 300, 320, 340, and 360 oC? Did the UN undergo thermal degradation? Thus TGA was suggested to provide in the revised manuscript.
- TGA data below (Figure R1) were added to supplementary information (Figure S2). The weight loss between 300 ℃ and 400 ℃ corresponds to the weight fraction of the aliphatic moiety of the UN. We added discussion on this in the revision (lines 144~146).
(3) what's the density and porosity of the carbon monolith? Could they compressible?
- The density was determined using the mercury intrusion porosimetry method. The bulk density of CRUN 340 was 0.53 g/mL. We noted this in the revision (line 250)
- The mesopore volume ratio over the total volume of the carbon monolith was calculated. We added data in Table S2 and noted this in the revision (line 208).
- The carbon was a rigid monolith and could not be compressed.
(4) Figure 1 must be revised, there is a vertical line in the left.
- Thank you. It was revised.

Round 2
Reviewer 2 Report
The authors have responsed to all the questions and now it can be accepted with no further changes.